# Secure and efficient graduate employment: A consortium blockchain framework with InterPlanetary file system for privacy-preserving resume management and efficient talent-employer matching

Chin-Ling Chen[1,2], Kuang-Wei Zeng[3], Hsing-Chung Chen[4,5]*, Yong-Yuan Deng[2], Chin-Feng Lee[3], Der-Chen Huang[6], Ling-Chun Liu[6]*

**1** School of Information Engineering, Changchun Sci-Tech University, Changchun, Jilin Province, China, **2** Department of Computer Science and Information Engineering, Chaoyang University of Technology, Taichung City, Taiwan, **3** Department of Information Management, Chaoyang University of Technology, Taichung City, Taiwan, **4** Department of Computer Science and Information Engineering, Asia University, Taichung City, Taiwan, **5** Department of Medical Research, China Medical University Hospital, China Medical University, Taichung City, Taiwan, **6** Department of Computer Information and Network Engineering & Master Program, Lunghwa University of Science and Technology, Taoyuan City, Taiwan

* cdma2000@asia.edu.tw (H-CC); 0287yell@gmail.com (L-CL)

## Abstract

In recent years, the unemployment situation of teenagers has become increasingly serious, and many college students face the problem of unemployment upon graduation. Concurrently, Companies need more support in their talent acquisition processes, including high costs, security concerns, inefficiencies, and time-consuming sourcing procedures. Moreover, job applicants frequently confront risks associated with potentially compromising their personal information during the application process. Since blockchain technology has the characteristics of non-tampering, traceability, and non-repudiation, it has outstanding significance for solving the trust problem between organizations. Blockchain has emerged as a powerful tool for tackling talent acquisition campaigns. This study proposes a novel approach utilizing consortium chain technology in conjunction with the InterPlanetary File System (IPFS) to develop a decentralized talent recruitment system. This approach enables students, educational institutions, and potential employers to encrypt and upload data to the blockchain through consortium chain technology, with strict access controls requiring student authorization for resume data retrieval. The proposed system facilitates a symbiotic relationship between educational institutions and industry partners, allowing students to identify suitable employment opportunities while enabling companies to source candidates with requisite expertise efficiently. Finally, the system could meet the characteristic requirements of various blockchains, perform well in terms of communication cost, computing cost, throughput, and transaction delay in the blockchain, and contribute to solving talent recruitment.

**Data availability statement:** All relevant data are within the paper, and this study does not have any relevant sources of information.

**Funding:** This work was supported by the Chelpis Quantum Tech Co., Ltd., Taiwan, under the Grant number of Asia University: I112IB120. This work was supported by the National Science and Technology Council (NSTC), Taiwan, under NSTC Grant numbers: 112-2410-H-324 -001 -MY2, 111-2218-E-468-001-MBK, 110-2218-E-468-001-MBK, 110-2221-E-468-007, 111-2218-E-002-037 and 110-2218-E-002-044. The funders had no role in the study design, data collection, and analysis, the decision to publish, or preparation of the manuscript.

**Competing interests:** The authors have declared that no competing interests exist.

## Introduction

The COVID-19 pandemic has significantly disrupted the global job market, severely affecting young adults. According to the International Labour Office (ILO), the global unemployment rate for individuals aged 18–24 reached 15.6% in 2021, affecting approximately 75 million young people. Notably, this rate was three times higher than that of adults [1]. This trend is especially concerning for college students facing unemployment upon graduation. The impact of unemployment extends far beyond mere economic hardship; it profoundly affects an individual's psychosocial needs, social participation, status, self-efficacy, sense of social integration, and overall well-being and mental health [2]. Consequently, unemployment significantly increases the risk of social exclusion, potentially leading to long-term negative outcomes for both individuals and society at large.

Pertinent to the challenges faced by graduating university students in the job market, it is crucial to consider the types of questions they may encounter during interviews. These often encompass inquiries about their academic achievements, internship experiences, career aspirations, and ability to apply theoretical knowledge to practical situations. Additionally, interviewers frequently probe candidates' soft skills, such as communication, teamwork, and problem-solving abilities. With this context in mind, it is worth examining the integrity issues that arise during the recruitment process [3]. Second, when companies recruit talents, job seekers include false information in their resumes to make themselves more attractive candidates [4,5]. In a 2021 analysis of 2.6 million Automatic Data Processing (ADP), Jasmine found that 23% had forged certificates or licenses, and 41% had lied on their resumes or falsified academic qualifications [6] With the development of the network, there are gradually more online talent recruitment systems, but it also creates a lot of data leakage risks [7,8] Statista counted the number of data breaches from the first quarter of 2020 to the third quarter of 2022, and in the fourth quarter of 2020, there were about 125 million data breaches on the Internet worldwide [9]. There are many recruitment scams when students are looking for a job. Recruitment scams range from using real jobs as bait to lure job seekers into sending passports, bank accounts, etc., personal information and paying unreasonable fees [10] to luring jobs into obtaining their names, phone numbers, and addresses through false job advertisements and then selling them to third parties [11]. Talent recruitment is primarily manual [12]. Recruitment is costly, unsafe, inefficient, and time-consuming [13]. Manual recruitment will also be abandoned in the next few years, turning to a safer and more efficient digital recruitment process [14].

With the rapid development of technology, more and more new technologies are being used to solve the problem of talent recruitment, and blockchain technology is one of them [15]. Blockchain technology represents a decentralized, pseudonymous, traceable, and non-repudiable information-sharing platform [16,17], This innovative technology demonstrates considerable scalability and robust security attributes, imbuing it with significant potential for deployment across a wide spectrum of industries and applications. Moreover, the inherent characteristics of blockchain technology offer promising solutions to address prevalent issues in talent recruitment, such

as fraudulent hiring practices and data breaches. Its versatility extends to addressing multifaceted challenges, including data privacy concerns [18–20], healthcare insurance paradigms [21], smart grid systems [22,23], secure data in the industrial Internet of Things [24,25], and smart environments [26]. In addition, other scholars have evaluated and improved the blockchain's consensus and incentive mechanism [27–29]. The architectural environment of blockchain can be classified into three types: public chain, private chain, and consortium chain. Although public chains are highly decentralized, they have low transaction speeds. Although the private chain has high data privacy and fast transaction speed, and the organization that enters the node needs permission to join, the authority is in the hands of a single organization. The organization running the private chain can easily modify the rules, resulting in low trust and greatly reduced security and transparency in the blockchain. The consortium blockchain combines the public chain's decentralization characteristics with the private chain's characteristics of high privacy and fast transaction speed and requires permission to join. Security and transparency in the blockchain will be significantly reduced. It is known that compared with public chains and private chains, consortium blockchain is more suitable for application in talent recruitment, an organization with multiple common interests.

At present, the use of blockchain technology to solve the talent recruitment system is mainly divided into the following two categories: (1) Discuss the design and application of recruitment: using blockchain technology digital certificate recruitment management platform [15], applicants can upload certificate documents to the blockchain for performance evaluation during the recruitment. An intelligent recruitment system for graduates based on big data-assisted blockchain is proposed [16], which uses big data technology to enable employers to recruit talents more efficiently. A disability recruitment model based on blockchain and smart contracts to change the lives of determined people with disabilities [30]. (2) Another category is to explore the protection of students' credits and certificates during the academic period: blockchain-based educational record storage and sharing scheme are proposed, enabling students to transfer credits between different educational institutions [10,31] Smart contracts are using Ethereum to enable cross-institutional sharing of education records [32]. There is a degree-proof system based on Hyperledger Fabric traceability [33].

However, in the above literature, the protection of credits and certificates during studies isn't combined with the recruitment process. It also doesn't use technologies to process data outside the blockchain to reduce the storage space on the blockchain. There's also no mention of data protection or information storage.

Therefore, this research aims to combine the consortium blockchain and IPFS with the talent recruitment system, allowing students, schools, and enterprises to utilize blockchain technology to ensure the privacy and security of the talent recruitment system. The objectives of this study are as follows:(1). Decentralization and Information Sharing: In the architecture of blockchain networks, decentralized bookkeeping and storage have been achieved, where each node enjoys equal rights and responsibilities. (2). Non-repudiation: Without non-repudiation, there is a possibility that the recruiters or applicants may falsify or tamper with the documents or agreements between them, resulting in a breach of trust or a loss of rights. (3). Data integrity: With data integrity, the company can accurately assess the recruitment needs when hiring. (4). Access control with privacy: There are many data breaches in online recruitment, so students can use this management system to open their resumes and learning experience files to enterprises according to their wishes to reduce the occurrence of data leakage. (5). Traceability: Blockchain technology ensures all parties have equal rights and choices, and any exchange between two roles is auditable. (6). Verifiability: Without verifiability, it is impossible to prove the authenticity or validity of data, transactions, or actions in the system, which leads to a lack of trust and credibility. (7). Man-in-the-Middle Attacks: Hackers may intercept network packets and modify or insert malicious code into messages that are being transmitted. (8). Sybil Attacks: Sybil attacks are behaviors in which individuals or organizations manipulate or control the system by creating multiple fake identities or nodes in the blockchain network.

The rest of the work is structured as follows. Section 2 provides the knowledge on which this study is based. Section 3 describes the talent acquisition management system's architecture and communication protocols. Section 4 is a security analysis of the system in this study. Section 5 discusses the computational cost, communication cost, and performance of the system and compares them with other schemes. In the last section, we conclude this work.

## 2 Preliminary

### 2.1 Blockcahin

The concept of blockchain technology was proposed by Satoshi Nakamoto in 2008 in the Bitcoin white paper [34]. Blockchain technology utilizes cryptography, consensus mechanisms, peer-to-peer network protocols, and other techniques to store transaction records in a decentralized database. The basic unit of a blockchain is a block, which contains transaction data, a timestamp, the block's hash value, and the previous block's hash value, as illustrated in Fig 1. Since each block contains the previous block's hash value, each block can be regarded as the fingerprint of the previous block, forming a continuously extending chain.

Blockchain includes many features, such as decentralization, transparency, anonymity, tamper resistance, traceability, scalability, encryption protection, and smart contracts.

### 2.2 Hyperledger Fabric

Hyperledger Fabric is an open-source, enterprise-grade, permissioned distributed ledger technology, a blockchain-based distributed ledger platform proposed by the Linux Foundation in 2015 [35]. The Hyperledger Fabric platform of the consortium blockchain must be approved for joining. This means that, unlike the public chain without permission, all consortium blockchain nodes must be vetted before joining. Unlike the full centralization of the private chain, it's jointly operated by several companies with stakeholders, which is particularly suitable for use in the enterprise environment. Hyperledger Fabric is a modular architecture containing Orderer, Peer, Client, Certificate Authorities (CA), Membership Service Providers (MSP), Channel, and Chaincode.

The basic Hyperledger Fabric architecture comprises a sorting node and multiple peer nodes. Before executing the chaincode, two nodes with common interests need to form a channel and submit the chaincode to the channel, and all users are connected to the Hyperledger server through the client; because the Hyperledger Fabric is a permission network, blockchain participants need a way to prove their identity to the rest of the network.

### 2.3 Smart Contract

Smart contracts were first proposed by Nick Szabo in 1994 [36]. In a smart contract, both parties agree on the details of the contract in advance, and when the pre-defined conditions are met, the program will be executed automatically.

Smart contract components: Smart contracts must have a contract subject so that the contract's relevant goods and service processes can be automatically locked and unlocked. Digital signature: Smart contracts require all participating users to authenticate through their private keys before they can be activated. Contract terms: All the operation procedures in the smart contract need to be recognized and signed by all participating users before they can be activated. The contract of the decentralized platform is deployed in the decentralized node platform, and we are waiting for the execution of the contract.

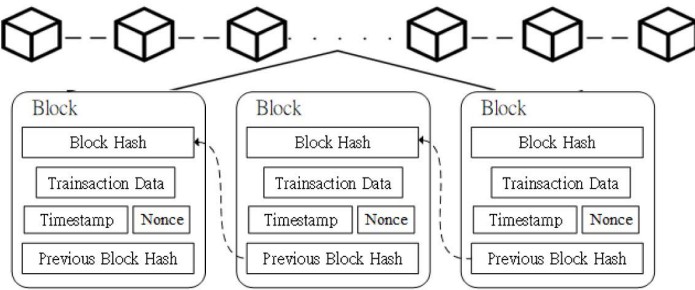

**Fig 1. Basic blockchain architecture.**

## 2.4 Interplanetary File System (IPFS)

The InterPlanetary File System (IPFS) is a distributed file storage and transmission system designed by Juan in 2014 [37]. By leveraging peer-to-peer technology and a decentralized approach, files are distributed and stored across nodes in the network, creating a more secure, faster, open, and sustainable network.

Utilizing the InterPlanetary File System (IPFS) for off-chain storage of voluminous data presents a cost-effective alternative to on-chain storage in blockchain systems. Upon transfer to IPFS, the data is subjected to SHA-256 cryptographic hashing, generating a distinct Content Identifier (CID). This process guarantees the uniqueness of each file within the IPFS network and ensures its immutability, thereby maintaining the integrity of the stored information.

## 2.5 Elliptic Curve Digital Signature Algorithm (ECDSA)

The Elliptic Curve Digital Signature Algorithm was proposed by Johnson et al. in 2001 [38], which is a combination of Elliptic Curve Cryptography (Elliptic Curve Cryptography, ECC) and Digital Signature Algorithm (DSA). Encryption algorithm, compared with RSA, ECDSA has a shorter public key length, smaller encrypted message, shorter calculation and processing time, and smaller memory and bandwidth requirements.

The three stages of ECDSA key generation are described as follows:

### 1) Key generation stage

Assuming that any participant must apply for public and private keys to the blockchain center, the key generation using ECDSA is as follows $Q_X = d_X G$, role x will be the participant's ID, $d_X$ the private key $Q_X$ the public key, and $G$ the generation point based on the ellipse curve. The public and private keys will be sent to the participant and stored, and the blockchain center will store them.

### 2) Signature Stage

First, you need to obtain a curve parameter $n$ and generate a random number $k$ between them, then calculate the hash value $h$ of the message $M$, and $h = Hash(M)$ then calculate a feature value on the curve $(x, y) = kG$. Then we calculate $r = x \bmod n$, and then we use the hash value of the message to sign: $s = k^{-1}(h + r * d_A) \bmod n$.

### 3) Verification stage

Verify that the values of $r$ and $s$, as well as $Q_X$ and $h$ match the following: $s = k^{-1}(h + r * d_A) \bmod n$, and then verify that the signature $r \overset{?}{=} x' \bmod n$ is valid.

## 3 Proposed Scheme

This chapter outlines the architecture and protocols of the proposed talent recruitment management system. Section 3.1 delineates the system's structural framework, detailing roles, systems, and organizations involved, and presents the process flow from the job posting to the interview stage. Section 3.2 defines the symbols and notations used. Sections 3.3–3.8 examine each phase of the recruitment process, detailing specific protocols and communication mechanisms employed at each stage, from job dissemination to interview actualization.

## 3.1 System Architecture

### 1) COMPANY (CO):

When a company intends to collaborate with a university for recruitment purposes, the company set up an internal corporate server and forms a consortium blockchain with the school, responsible for posting job offers to the school.

---

2) **STUDENT (STU):**

Students can search for the jobs they want from the job openings posted by the company, and authorize the school to allow the company to check the student's information.

3) **SCHOOL (SCH):**

Linking student's expertise system with the blockchain center, setting up a blockchain server to create a consortium blockchain channel and adding companies to the channel to form a consortium blockchain, and then the school administrator to build a bridge for communication between students and companies, allowing students to authorize the school.

4) **CERTIFICATE AUTHORITY (CA):**

CA node will provide certificates, public keys, and private keys for all users who want to join the system.

5) **BLOCKCHAIN CENTER (BCC):**

The school administrator sets up Blockchain Center and contains fabric peer nodes (school and company) and fabric Orderer nodes and Certificate Authority (CA) nodes deployed using Docker technology.

6) **DOCKER:**

Using Docker to deploy Hyperledger Fabric can improve deployment efficiency and provide modularity, virtualization, and flexibility. Additionally, it provides a secure isolated container environment to protect the system security and is easy to manage and maintain.

7) **INTERPLANETARY FILE SYSTEM (IPFS):**

Schools can upload students' academic history to the blockchain center, such as resumes, academic history, personal academic achievement, and professional licenses obtained.

Fig 2 illustrates the complete process from the publication of job recruitment information to student-company interviews, involving students, schools, and companies. The detailed description of the six stages is as follows:

1) **Initialization phase:**

The school administrator will first create a channel, then add the peer nodes of the company and the school to the channel, install the chaincode on each peer node, and initialize the chaincode.

2) **Registration phase:**

All users (including students and companies) need to apply for registration with the blockchain center to get the system account.

3) **Company recruitment announcement phase:**

When the company has job vacancies, it will propose the list of required professionals to the school, the conditions, and the deadline for the candidates.

4) **Student job application phase:**

Students can apply for an interview on the website provided by the school within the deadline and fill out the interview information. After the deadline, the school will judge whether the students conform to the list of expertise, and then upload the interview information and learning history file to IPFS after encrypting the information that meets the list of expertise, and getting the address of the information in IPFS, and then store it into the blockchain system with the student number.

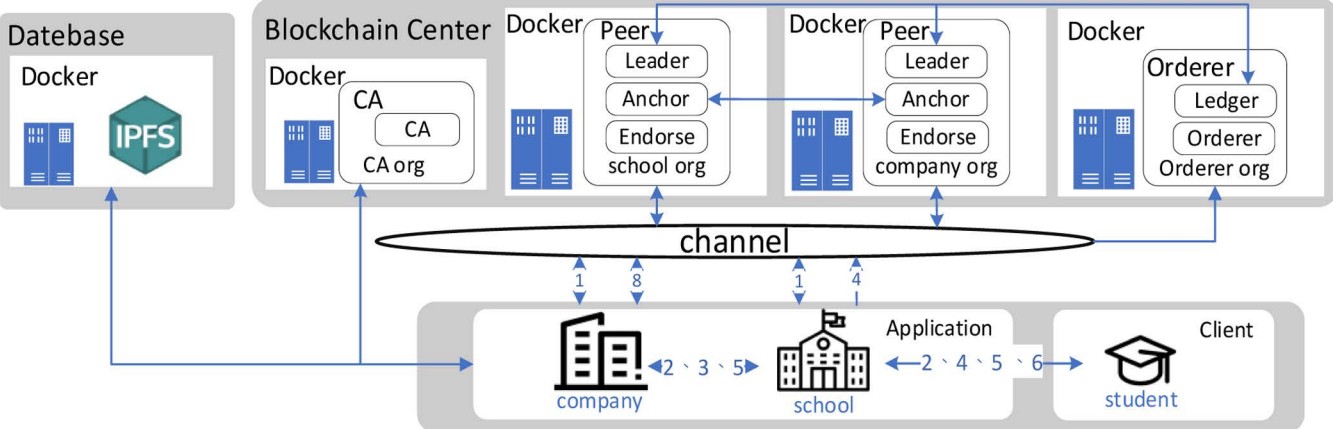

**Fig 2. System architecture diagram.**

## 5) Interview phase:

The school will send the authorization request message to the student. When the student receives the message, he/she can choose whether or not to authorize the school to reveal the student's data for the company to view the student's data, and platform to announce the interview. The school will send a list of eligible students with their student numbers and a one-time URL to the company. The company will use the student list and one-time password to check the student's interview information and learning history and can make further interviews with the qualified students.

## 6) Announcement results phase:

The company will send the information to the students whether they passed the interview or not.

### 3.2 Notation

The notation of this article is illustrated in Table 1.

### 3.3 Initialization Phase

During the channel creation phase, the school administrator sets up a Hyperledger Fabric platform and organizes the companies and schools into a consortium blockchain. The school administrator creates a channel and then initializes the chain by adding each node of the corresponding partner organizations (companies and schools) to the channel and installing the chaincode.

Step 1: The school administrator will create the necessary files for the channel and use the Application to create the channel and add the school and partner companies to the channel.

Step 2: The school administrator connects all peer nodes in the channel through the Application then installs the chain-code on each peer node and initializes the chaincode.

Figs 3 and 4 show the chaincode structure and IPFS structure, respectively, for storing job announcement messages and job applications. Before storing the job announcement messages and job applications in the blockchain, they have to be encrypted and stored in IPFS. Then the index addresses are added with the corresponding company_id, deadline, or student_id before they can be stored in the blockchain.3.4 Registration Phase.

**Table 1. Notations.**

| $M_i$ | The $i$-th Message from the sender. |
|---|---|
| $M_{Auth}$ | The authorization request message is sent by the school to the students. |
| $ID_X$ | The identity of character $X$. |
| $CV$ | Interview data of the student |
| $ID_{CV}$ | The ID of the student's interview data. |
| $ID_{List}$ | The list of students who meet the competencies proposed by the company. |
| $specialties$ | The list of competencies proposed by the company. |
| $e$-$portfolio$ | File with the student's academic history. |
| $Hash()$ | One-way hash function. |
| $H_{X_i}$ | The $i$-th hash of role $X$. |
| $k_{X_i}$ | The $i$-th hash of role $X$ generates a random number on the elliptic curve. |
| $C_{X_i}$ | The $i$-th ciphertext of role $X$. |
| $E$ | The elliptic curve is defined on a finite group. |
| $G$ | Generation point based on an elliptic curve $E$. |
| $Q_X/d_X$ | Public/private key of role $X$. |
| $Cert_X$ | Digital certificate of role $X$ conforming to the X.509 standard. |
| $(x_i, y_i)$ | The $i$-th elliptic feature value. |
| $(r_i, s_i)$ | The $i$-th ECDSA signature value. |
| $E_{Puk_X}(M)/D_{Prk_X}(C)$ | Use role $X's$ public key $Puk_x$ to encrypt the message or use role $X$ private key $Prk_x$ to decrypt ciphertext $C$. |
| $x \overset{?}{=} r$ | Verify whether $x$ is equal to $r$. |
| $OTP$ | The threshold for checking the validity of timestamps. |
| $CID_{CO}$ | The CID used to publish a job posting on IPFS Query Company. |
| $CID_{STU}$ | The CID used to query student files on IPFS. |

```
type Job_opening_ipfs struct{          type Job_application_ipfs struct{
  Company_id string `json:"company_id"`   Student_id string `json:"student_id"`
  Specialties string `json:"specialties"`  Curriculum_vitae string `json:"curriculum_vitae"`
}                                         E-portfolio string `json:"e-portfolio"`
                                         }
```

**Fig 3. IPFS structure for job search announcements and job applications.**

```
type POA struct{                           type Job_opening struct{
  Student_id  string `json:"student_id"`     Company_id string `json:"company_id"`      type Job_application struct{
  Student_sign string `json:"student_sign"`  IPFS_address string `json:"ipfs_address"`    IDCV string `json:"idcv"`
  IDCV string `json:"idcv"`                  Company_sign string `json:"company_sign"`    Student_sign string `json:"student_sign"`
  Company_sign string `json:"company_sign"`  Deadline string `json:"deadline"`            IPFS_address string `json:"ipfs_address"`
}                                          }                                          }
```

**Fig 4. Chaincode structure of job search announcements and job applications.**

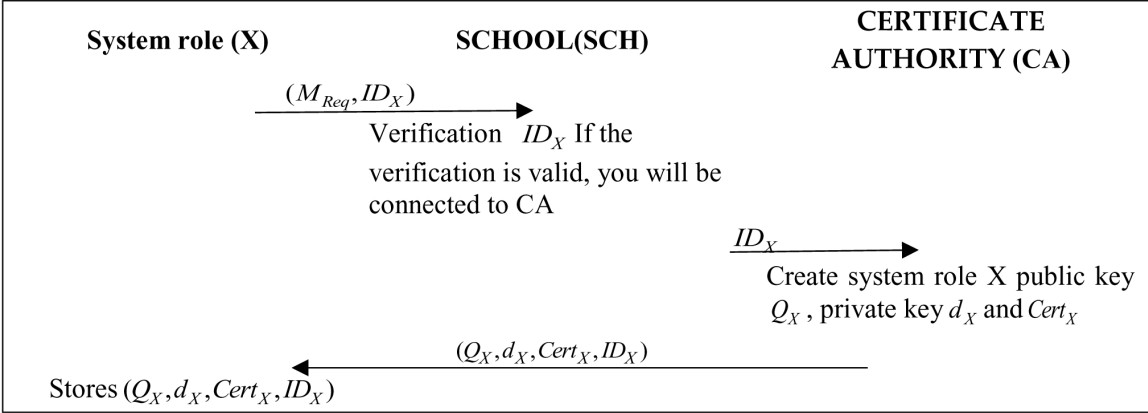

**Fig 5. Protocol of the registration phase.**

During the registration phase, system role X can represent both companies and students. Role X submits enrollment application requirements and identity information to the school to the CA node. The CA generates the public key, and certificate for Role X. Fig 5 shows a flowchart of the user registration phase.

Step 1: System role X generates a registration application request message, and the identity code is sent to the school.

Step 2: The SCH administrator will verify the information and, if valid, send it to the corresponding CA node, which will generate the public key, private key, and certificate document based on the system's role.

Step 3: The system will send the CA-generated information to system role X based on system role X's account and identity information.

Step 4: System role X receives the parameter of the signed message and saves it.

### 3.5 Company Recruitment Announcement Phase

When a company has a job opening, it will submit the job requirements, a list of required skills, and the deadline to the school through the website, along with an ECDSA signature to ensure that the request is made by the company. This is the job posting stage, as illustrated in Fig 6.

Step 1:When a company has a job requirement, the company will generate a job requirement $M_{Jo}$, $ID_{CO}$, a requirement specialty *specialties*, a random number $k_{CO_1}$ and a *deadline* and then generate a message $M_{Jo}$, $ID_{CO}$ and *specialties* into $M_1$:

$$M_1 = (ID_{CO}||M_{Jo}||specialties) \qquad (1)$$

$$H_{CO_1} = hash(M_1) \qquad (2)$$

The company will then digitally sign the *Sign()* function in the $(H_{CO_1}, k_{CO_1}, d_{CO})$ parameter transfer Algorithm1. $(r_1, s_1)$ and then the company will send $(ID_{CO}, M_1, H_{CO_1}, (r_1, s_1), deadline)$ to the school administrator.

Step 2: The school administrator then performs hashing function by using $M_1$ and verifying if the hash value $M_1$ is the same:

$$H'_{CO_1} = hash(M_1) \qquad (3)$$

$$H_{CO_1} \stackrel{?}{=} H'_{CO_1} \qquad (4)$$

The school administrator will then call the *Verify()* function of Algorithm 2 with the parameter $(H'_{CO_1}, r_1, s_1, Q_{CO})$ to verify signature is correct. If the school administrator verifies that the signature is valid send $M_1$ it to IPFS.

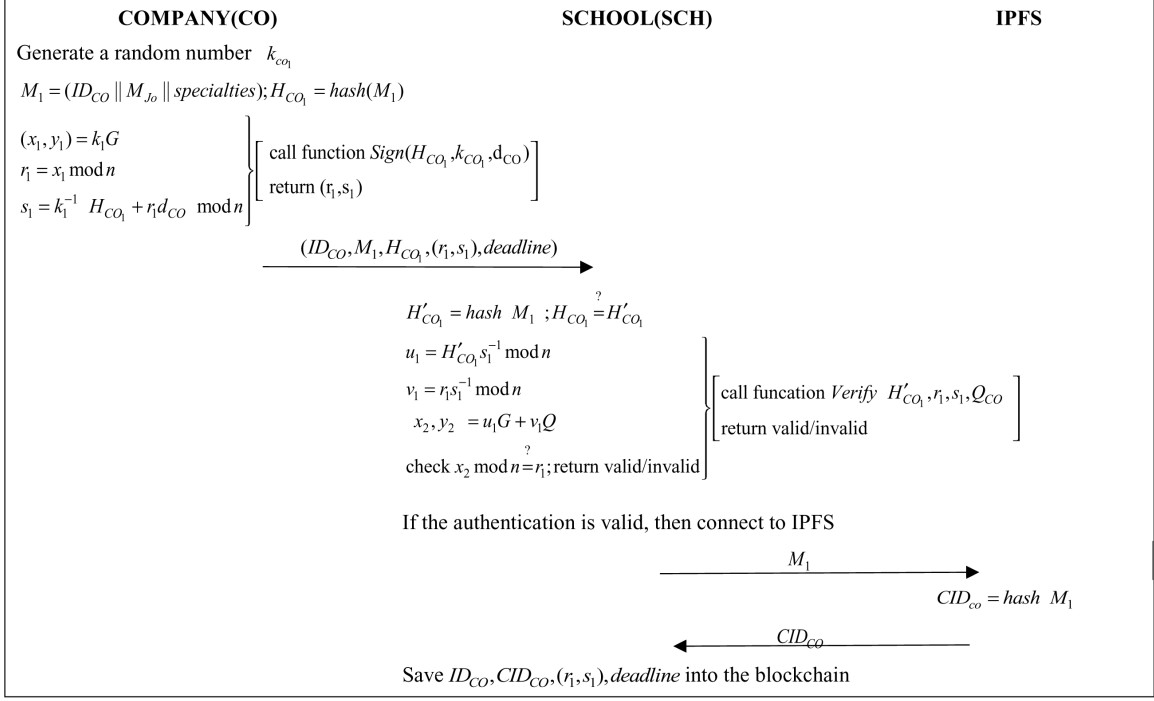

**Fig 6. Protocol of company recruitment announcement phase.**

Step 3: IPFS calculation query code $CID_{CO}$:

$$CID_{co} = hash(M_1) \tag{5}$$

IPFS sends $CID_{CO}$ to the school administrator.

Step 4: The school administrator then saves $ID_{CO}$, $CID_{CO}$, $(r_1, s_1)$, and into the blockchain.

---

**Algorithm 1.** *function sign$((h, k, d))$*

$(x, y) = k * G$
$r = x \bmod n$
$s = k^{-1}(h + r * d) \bmod n$
$return(r, s)$

---

---

**Algorithm 2.** *function Verify$((h, r, s, Q))$*

$u_1 = h * s^{-1} \bmod n$
$v_1 = r * s^{-1} \bmod n$
$(x, y) = u_1 G + v_1 Q$
$return((x \bmod n == r)?''valid'':''invalid'')$

---

## 3.6 Student Job Application Phase

In the job application process, students submit requests via the institution's portal, providing interview details, credentials, and employer designation. Requests include an ECDSA signature for authenticity. Upon meeting the criteria, student

records and information undergo a two-phase process: uploading to IPFS, generating a Content Identifier (CID), and then recording on the blockchain, as shown in Fig 7.

Step 1: Students fill out the interview data at $CV$ a random number $k_{STU_1}$ and $ID_{STU}$ and students use $CV$, $ID_{STU}$ and $ID_{CO}$ to generate the message $M_2$ and perform hashing operation:

$$M_2 = (ID_{STU}||ID_{CO}||CV) \tag{6}$$

$$H_{STU_1} = hash\,(M_2) \tag{7}$$

The student will then obtain a digital signature for the $Sign()$ function in the $(H_{STU_1}, k_{STU_1}, d_{STU})$ parameter delivery Algorithm 1 $(r_2, s_2)$ and then encrypt $M_2$ using the school's public key and generate the ciphertext $C_{STU_1}$:

$$C_{STU_1} = E_{Puk_{SCH}}(M_2) \tag{8}$$

Finally, the student sends $(ID_{STU}, C_{STU_1}, H_{STU_1}, (r_2, s_2))$ to the school for signature verification.

Step 2: After receiving the message, the school administrator will use the school's private key to decrypt the ciphertext $C_{STU_1}$ and perform a hashing function to verify if the hash count $M_2$ is the same:

$$M_2 = D_{Prk_{SCH}}(C_{STU_1}) \tag{9}$$

$$H'_{STU_1} = hash\,(M_2) \tag{10}$$

$$H_{STU_1} \overset{?}{=} H'_{STU_1} \tag{11}$$

Next, the school administrator will verify that the $Verify()$ function in the $(H'_{STU_1}, r_2, s_2, Q_{STU})$ parameter Algorithm 2 is signed correctly. If the signature is valid the school administrator will query the student's $e$-portfolio, and then use $ID_{STU}$, $ID_{CO}$, $CV$ and $e$-portfolio to generate a message $M_3$, using the school's public key to generate $C_{SCH_1}$:

$$M_3 = (ID_{STU}||ID_{CO}||CV||e\text{-}portfolio) \tag{12}$$

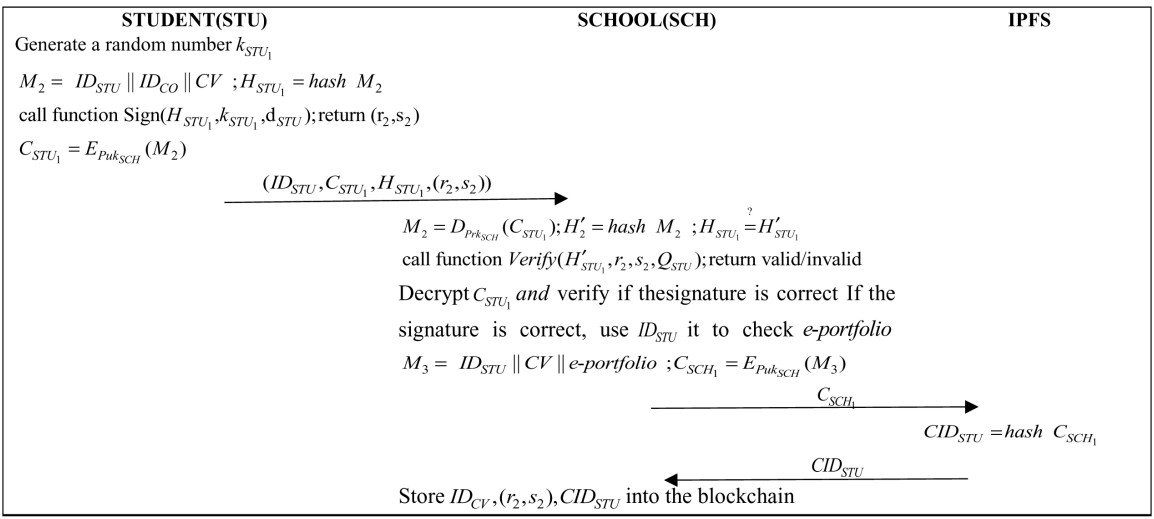

**Fig 7. Protocol of student job application phase.**

$$C_{SCH_1} = E_{Puk_{SCH}}(M_3) \tag{13}$$

The school administrator will then store the encrypted cipher text $C_{SCH_1}$ in IPFS.
Step 3: IPFS calculates the query code $CID_{STU}$:

$$CID_{STU} = hash\,(C_{SCH_1}) \tag{14}$$

IPFS sends $CID_{STU}$ to the school administrator.
Step 4: The school then generates $ID_{CV}$ and stores $ID_{CV}$, $CID_{STU}$ and $(r_2, s_2)$ into the blockchain.

### 3.7 Interview Phase

The school will send a request for authorization message to the student when the deadline for the job announcement is reached. Upon receipt of the request, the student can decide whether or not to authorize the request and authorize the school with an ECDSA signature, allowing the school to open the student's information to the company for inquiries. The school sends the list of eligible students to the company with their one-time authorization code. The company will then use the student list and the one-time authorization password to access the student's interview information and learning history and can schedule further interviews with eligible students, as illustrated in Figs 8 and 9.
Step 1: The school administrator will send an authorization request to $M_{Auth}$ the student.
Step 2: Students generate a random number $k_{STU_2}$ and then use $ID_{STU}$, $ID_{SCH}$, $ID_{CO}$ and $ID_{CV}$ to generate the message $M_4$ and perform the hashing operation:

$$M_4 = (ID_{STU}||ID_{SCH}||ID_{CO}||ID_{CV}) \tag{15}$$

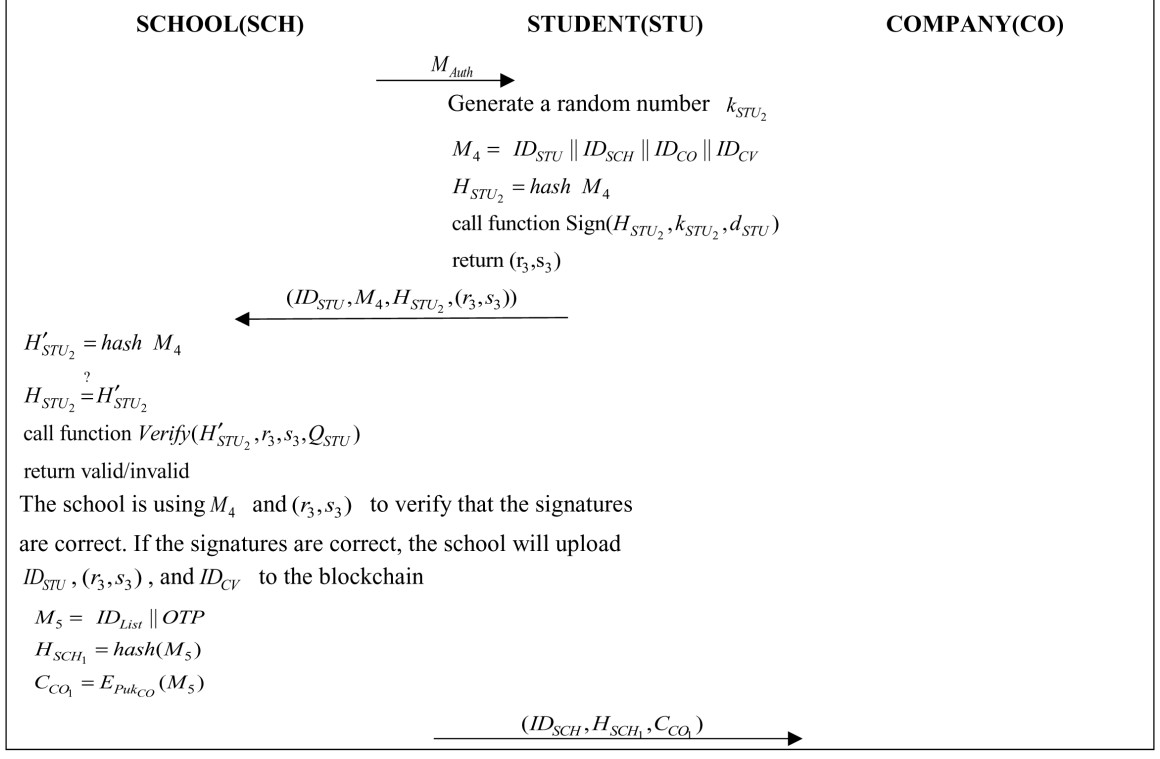

**Fig 8. Protocol of interview phase 1.**

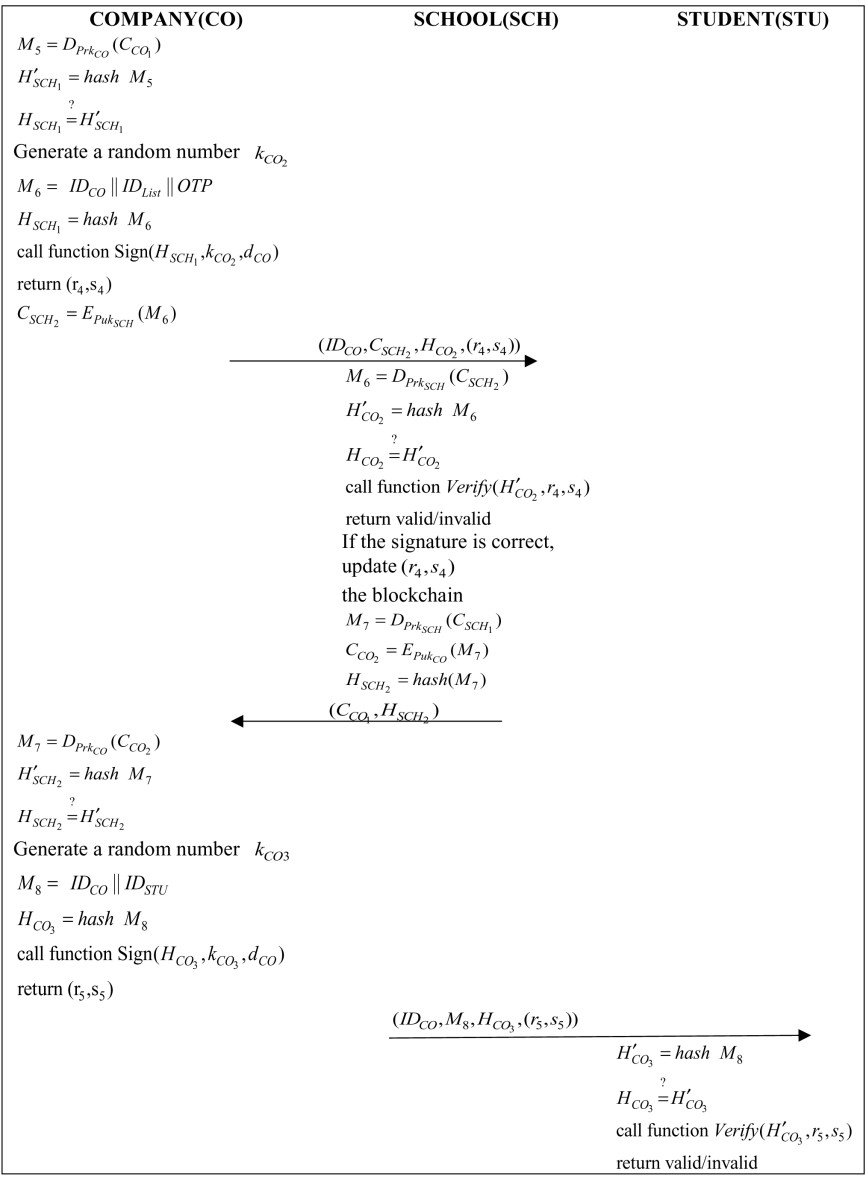

**Fig 9. Protocol of interview phase 2.**

$$H_{STU_2} = hash\,(M_4) \tag{16}$$

The student then passes the $(H_{STU_2}, k_{STU_2}, d_{STU})$ parameter to the Sign function in Algorithm 1 to obtain a digital signature $(r_3, s_3)$ and the student then passes $(ID_{STU}, M_4, H_{STU_2}, (r_3, s_3))$ it to the school.

Step 3: The school administrator will use $M_4$ for hashing operation and will verify if the hash value $M_4$ is the same as $H_{STU_2}$ illustrated below.

$$H'_{STU_2} = hash\,(M_4) \tag{17}$$

$$H_{STU_2} \stackrel{?}{=} H'_{STU_2} \tag{18}$$

The school administrator will pass the parameters $(H'_{STU_2}, r_3, s_3, Q_{STU})$ to the Verify function in Algorithm 2 to confirm whether the signature is correct. After confirming the signature, the school administrator will upload $ID_{STU}$, $(r_3, s_3)$ and $ID_{CV}$ to the blockchain using the chaincode, and then the school will generate a message $ID_{List}$ with the one-time authorization code and generate the message $M_5$ using $ID_{List}$ and $OTP$, finally encrypt it using the company's public key:

$$M_5 = (ID_{List}||OTP) \tag{19}$$

$$H_{SCH_1} = hash(M_5) \tag{20}$$

$$C_{CO_1} = E_{Puk_{CO}}(M_5) \tag{21}$$

The school administrator then sends $(ID_{SCH}, H_{SCH_1}, C_{CO_1})$ to the company.

Step 4: The company decrypts the ciphertext $C_{CO_1}$ with its private key, then the company performs a hashing operation and verifies if the hash value $M_5$ is the same as $H_{SCH_1}$:

$$M_5 = D_{Prk_{CO}}(C_{CO_1}) \tag{22}$$

$$H'_{SCH_1} = hash(M_5) \tag{23}$$

$$H_{SCH_1} \stackrel{?}{=} H'_{SCH_1} \tag{24}$$

The company generates a random number $k_{CO2}$ and uses $ID_{CO}$, $ID_{List}$ and $ID_{Url}$ to generate and hash the message $M_6$:

$$M_6 = (ID_{CO}||ID_{List}||OTP) \tag{25}$$

$$H_{CO_2} = hash(M_6) \tag{26}$$

The company will then pass the $(H_{CO_2}, k_{CO_2}, d_{CO})$ parameter to the Sign function in Algorithm 1 to obtain the digital signature $(r_4, s_4)$ and encrypt it with the school's public key, and then the company will generate the cipher text $C_{SCH_2}$:

$$C_{SCH_2} = E_{Puk_{SCH}}(M_6) \tag{27}$$

Finally, the company sends $(ID_{CO}, C_{SCH}, H_{CO_2}, (r_4, s_4))$ to the school.

Step 5: The school administrator will decrypt the ciphertext $C_{SCH_2}$ with his private key and perform a hashing operation and hash value verification:

$$M_6 = D_{Prk_{SCH}}(C_{SCH_2}) \tag{28}$$

$$H'_{CO_2} = hash(M_6) \tag{29}$$

$$H_{CO_2} \stackrel{?}{=} H'_{CO_2} \tag{30}$$

The school administrator will pass the $(H'_{CO_2}, r_4, s_4)$ parameter to the Verify function in Algorithm 2 to verify if the signature is correct. After verifying the signature, the school administrator will use $ID_{List}$ to find the student $ID_{CV}$ and use $ID_{CV}$

to find $key_{STU}$. Finally, use $key_{STU}$ to find the ciphertext $C_{SCH_1}$ in IPFS and then use the school's private key to decrypt the ciphertext $C_{SCH_1}$:

$$M_7 = D_{Prk_{SCH}}(C_{SCH_1}) \tag{31}$$

The signature message is then updated to the blockchain using the chaincode and encrypted using the company's public key:

$$C_{CO_2} = E_{Puk_{CO}}(M_7) \tag{32}$$

$$H_{SCH_2} = hash(M_7) \tag{33}$$

Finally, the school administrator sends $(C_{CO_1}, H_{SCH_2})$ to the company.

Step 6: The company will use its private key to decrypt the ciphertext at $C_{CO_2}$:

$$M_7 = D_{Prk_{CO}}(C_{CO_2}) \tag{34}$$

$$H'_{SCH_2} = hash(M_7) \tag{35}$$

$$H_{SCH_2} \overset{?}{=} H'_{SCH_2} \tag{36}$$

After decryption, the company can view the students $CV$ and *e-portfolio* can schedule further interviews with eligible students. To schedule further interviews, the company generates a random number $k_{CO_3}$ and uses $ID_{CO}$ and $ID_{STU}$ to generate the message $M_8$ and perform hashing operation:

$$M_8 = (ID_{CO}||ID_{STU}) \tag{37}$$

$$H_{CO_3} = hash(M_8) \tag{38}$$

Step 7:The student will use $M_8$ to perform the hashing operation and verify if the hash value $M_8$ is the same:

$$H'_{CO_3} = hash(M_8) \tag{39}$$

$$H_{CO_3} \overset{?}{=} H'_{CO_3} \tag{40}$$

The student will pass the $(H'_{CO_3}, r_5, s_5)$ parameters to the *Verify()* function in Algorithm 2 to verify if the signature is correct, and then the student can wait for the company to conduct further oral interviews.

## 3.8 Announcement Results Phase

After the company has completed the interview, a list of those who have passed the interview will be sent to the school with the company's signature to ensure that the message was sent by the company and then posted on the announcement board. Fig 10 shows the flow of the announcement of results.

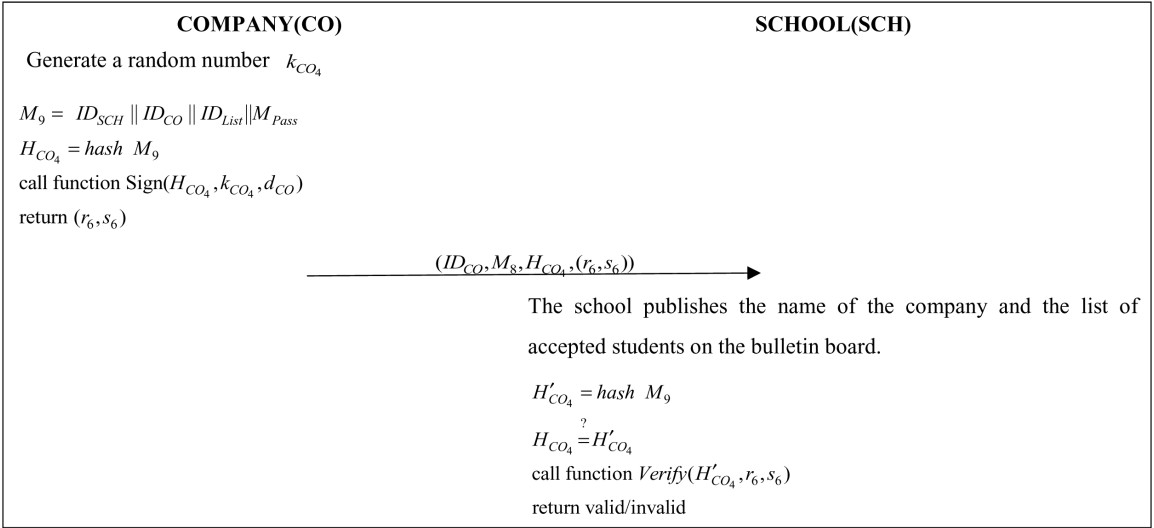

**Fig 10. Protocol of announcement results phase.**

Step 1: The company generates random numbers $k_{CO_4}$ and uses $ID_{STU}$, $ID_{SCH}$ and the messages that pass the interview $M_{Pass}$ to generate messages and perform hashing operation:

$$M_9 = (ID_{SCH}||ID_{CO}||ID_{List}||M_{Pass}) \tag{41}$$

$$H_{CO_4} = hash\,(M_9) \tag{42}$$

The company will send the $(H_{CO_4}, k_{CO_4}, d_{CO})$ parameters to the Sign function in Algorithm 1 to obtain the digital signature $(r_6, s_6)$ and then send $(ID_{CO}, M_9, H_{CO_4}, (r_6, s_6))$ to the school.

Step 2: The school administrator will perform hash computation on $M_9$ and check if the hash value $H'_{CO_4}$ of $M_9$ is the same as $H_{CO_4}$:

$$H'_{CO_4} = hash\,(M_9) \tag{43}$$

$$H_{CO_4} \overset{?}{=} H'_{CO_4} \tag{44}$$

The school administrator will pass the $(H'_{CO_4}, r_6, s_6)$ parameter to the *Verify*() function in Algorithm 2 to verify if the signature is correct. If it is, post the company name and list of accepted students in the announcement section

## 4 Analysis

In the following sections, this study presents a comprehensive analysis of blockchain technology, encompassing its fundamental characteristics, security assessments, and an examination of common network threats and security challenges blockchain systems face. We begin with a detailed analysis of the core characteristics of blockchain (Sections 4.1 to 4.4). Subsequently, we conduct a thorough security assessment (Sections 4.5 and 4.6). Finally, we explore common network threats and security challenges blockchain systems face (Sections 4.7 and 4.8).

## 4.1 Decentralization and Information Sharing

Each role handles the data and uses his or her private key to sign in the proposed scheme. We have adopted Hyperledger's consortium chain architecture. Under the Hyperledger Fabric, consortium members must register with the blockchain to communicate. The Hyperledger Fabric architecture allows for forming federations of interested players, and a single Hyperledger Fabric architecture can have multiple federations. All information dissemination is transparent and open to members of the same federation. Due to the blockchain architecture, any one node cannot cheat the rest of the nodes. For the members of the same federation, this realizes the trust relationship among the members and further constitutes the trust relationship among multiple members. Therefore, the proposed method realizes federation members' decentralization and information sharing under the Hyperledger Fabric architecture of blockchain.

## 4.2 Traceability

After the information is uploaded to the blockchain, the data blocks containing the transaction information will be permanently stored on the blockchain and cannot be tampered with. For example, when you want to track and confirm the legality of the blockchain data between the company and the school during the job search announcement stage, you can compare and verify the signature value, and when you want to verify the legality of the blockchain between the student and the school during the job search application stage, you can compare and verify.

## 4.3 Data Integrity

To ensure the integrity of the data in the sender and receiver communications and the data in the memory, this scenario uses the hash values of the sender and receiver for evaluation during the data-sending phase, as illustrated in Table 2. In the storage phase, IPFS generates a unique search code for each record and finally stores it in the blockchain. This way, hackers cannot falsify the data, and the integrity of the data is guaranteed.

## 4.4 Non-repudiation

The surname is indisputable, and when the recipient receives the message, he verifies it with the sender's public key. If the message is successfully verified, the sender cannot deny the transmitted message. The information is then uploaded

**Table 2. Verification of integrity.**

| Stage | Party | | Data | Hash Value | | | Verify |
|---|---|---|---|---|---|---|---|
| | Sender | Receiver | | | | | |
| Company recruitment announcement stage | CO | SCH | $M_1 = ID_{CO}, M_{Jo}, specialties$ | $H_{CO_1} = hash(M_1)$ | $H'_{CO_1} = hash(M_1)$ | | $H_{CO_1} \overset{?}{=} H'_{CO_1}$ |
| Student job application stage | STU | SCH | $M_2 = ID_{STU}, ID_{CO}, CV, M_{App}$ | $H_{STU_1} = hash(M_2)$ | $H'_{STU_1} = hash(M_2)$ | | $H_{STU_1} \overset{?}{=} H'_{STU_1}$ |
| Interview phase | STU | SCH | $M_4 = ID_{STU}, ID_{SCH}, ID_{CO}, ID_{CV}, M_{Agr}$ | $H_{STU_2} = hash(M_3)$ | $H'_{STU_2} = hash(M_4)$ | | $H_{STU_2} \overset{?}{=} H'_{STU_2}$ |
| | SCH | CO | $M_5 = ID_{List}, M_{Url}$ | $H_{SCH_1} = hash(M_4)$ | $H'_{SCH_1} = hash(M_5)$ | | $H_{SCH_1} \overset{?}{=} H'_{SCH_1}$ |
| | CO | SCH | $M_6 = ID_{CO}, ID_{List}, M_{Url}$ | $H_{CO_2} = hash(M_5)$ | $H'_{CO_2} = hash(M_6)$ | | $H_{CO_2} \overset{?}{=} H'_{CO_2}$ |
| | SCH | CO | $M_7 = D_{Prk_{SCH}}(C_{SCH_3})$ | $H_{STU_3} = hash(M_7)$ | $H'_{STU_2} = hash(M_7)$ | | $H_{SCH_2} \overset{?}{=} H'_{SCH_2}$ |
| | CO | STU | $M_8 = ID_{CO}, ID_{STU}$ | $H_{CO_3} = hash(M_8)$ | $H'_{CO_3} = hash(M_8)$ | | $H_{CO_3} \overset{?}{=} H'_{CO_3}$ |
| Announcement results phase | CO | SCH | $M_9 = ID_{SCH}, ID_{CO}, ID_{List}, M_{Pass}$ | $H_{CO_4} = hash(M_9)$ | $H'_{CO_4} = hash(M_9)$ | | $H_{CO_4} \overset{?}{=} H'_{CO_4}$ |

to a blockchain, and the timestamp function included in the blockchain records the time of creation. The non-repudiation verification is illustrated in Table 3.

## 4.5 Access Control with Privacy

In this scheme, the architecture based on the consortium chain is adopted, combined with the encryption of the public and private keys and the off-chain storage mechanism of IPFS. The message is signed by the private key, uploaded to IPFS, and then the query code is transmitted to the blockchain center, which can confirm the correctness of the information of all members of the same consortium and the openness of the blockchain system. In addition, the text of the message is also encrypted by the public key, and only the consortium members with the corresponding private key can decrypt the correct information, while the rest of the members cannot get the correct information to achieve the protection of private information. Through the Consortium Blockchain structure, the sharing of information and the protection of personal privacy can be realized at the same time.

## 4.6 Verifiable

This paper implements several verifiable goals in the system using blockchain architecture. For example, digital certificates can publicly verify the legitimacy of role identities, and each role's public key and signature information can also be made available to consortium members. According to the characteristics of the blockchain, the transaction process from the licensed user to the owner also needs to be uploaded to the blockchain, which will also become the object of public supervision.

When a company sends a signature message from ECDSA to a school, the school verifies the correctness of the signature by using $Verify()$ Algorithm 2, then generates the blockchain data $(r_1, s_1)$, and later uploads the blockchain data as an index to the blockchain, that is, after verifying the correctness of the incoming message and the signature of each role, it also verifies the correctness of all the blockchain data generated by the previous roles. This means that after confirming the correctness of incoming messages and signatures of each actor, the correctness of all blockchain data generated by previous roles is also verified. Thus, the solution realizes the feature of public verification using ECDSA digital signature and blockchain technology.

## 4.7 Man-in-the-middle Attack (MITM)

MITM is one of the most well-known cyberattacks [39], in which the attacker alters or intercepts the messages between the sender and the receiver. In this method, we use asymmetric encryption, e.g., students encrypt into a cipher with the school's public key during the application phase. When the school receives the encrypted text, they decrypt it with their private key and the equation is as follows:

$$C_{STU_1} = E_{Puk_{SCH}}(M_2) \tag{45}$$

$$M_2 = D_{Prk_{SCH}}(C_{STU_1}) \tag{46}$$

**Table 3. Verification of non-repudiation.**

| Stage | Signature | Party | | Verify |
|---|---|---|---|---|
| | | Sender | Receiver | |
| Company recruitment announcement phase | $(r_1, s_1) = Sign(H_{CO_1}, k_{CO_1}, d_{CO})$ | CO | SCH | $Verify(H'_{CO_1}, r_1, s_1, Q_{CO})$ |
| Student job application phase | $(r_2, s_2) = Sign(H_{STU_1}, k_{STU_1}, d_{STU})$ | STU | SCH | $Verify(H'_{STU_1}, r_2, s_2, Q_{STU})$ |
| Interview phase | $(r_3, s_3) = Sign(H_{STU_2}, k_{STU_2}, d_{STU})$ | STU | SCH | $Verify(H'_{STU_2}, r_3, s_3, Q_{STU})$ |
| | $(r_4, s_4) = Sign(H_{CO_2}, k_{CO_2}, d_{CO})$ | CO | SCH | $Verify(H'_{CO_2}, r_4, s_4, Q_{CO})$ |
| | $(r_5, s_5) = Sign(H_{CO_3}, k_{CO_3}, d_{CO})$ | CO | STU | $Verify(H'_{CO_3}, r_5, s_5, Q_{CO})$ |
| Announcement results phase | $(r_6, s_6) = Sign(H_{CO_4}, k_6, d_{CO})$ | CO | SCH | $Verify(H'_{CO_4}, r_6, s_6, Q_{CO})$ |

Scenario: The attacker uses the communication messages between the sender and the receiver to eavesdrop or modify the content and analyze the content.

Analysis: The message is encrypted with the recipient's public key and the recipient must have got a corresponding private key for decryption to obtain the message. However, since the attacker doesn't have the recipient's private key, he cannot decrypt, learn and forward the message.

### 4.8 Sybil Attack

Sybil attacks are attacks on P2P networks where attackers take control of the networks by impersonating multiple identities and breaking consensus mechanisms [40]. In this solution, our system uses the Hyperledger Fabric architecture, where all nodes that want to join must be authorized by a credential manager (CA) to join nodes, and all nodes are authenticated with certificates to avoid Sybil attacks.

## 5  Discussion

Section 5.1 of this chapter will analyze the computational costs for each phase. Section 5.2 will examine the communication costs for each phase, evaluating them in both Fast Ethernet and 10 Gigabit Ethernet environments. Section 5.3 will present performance tests based on the proposed solution, focusing on the performance evaluation of chaincode deployment. In the test, we selected 13 sending rate groups ranging from 50 tps to 650 tps, with 50 tps intervals between each group, each lasting 3 seconds, and employing load balancing techniques. Furthermore, we will analyze the relationship between sending rate and latency, as well as the relationship between sending rate and throughput. Finally, we will analyze the access capability and throughput of the proposed system. Section 5.4 will compare our approach with previous research using a tabular format.

### 5.1  Computation Cost

The computational costs for each phase are analyzed in Table 4. We use asymmetric encryption/decryption, comparisons, hash functions, and multiplication calculations as the basis for the computational cost. Calculated Costs per Phase

### 5.2  Communication Cost

The communication costs are analyzed for each phase in Table 5. In a Fast Ethernet environment, the maximum transmission rate is 100Mbps, and in a 10 Gigabit Ethernet environment, the maximum transmission rate is 10 Gbps. From the analysis in Table 5, it could be seen that the proposed solution has excellent performance in both high-speed Ethernet and exile Ethernet environments.

### 5.3  Performance

For the scenario proposed in this study, this section evaluates the performance of the chaincode contract deployment. The test tool uses Hyperledger Calliper version 0.4.2 and the blockchain platform uses Hyperledger Fabric version 2.3. On a server with an Intel Core i9 9920X@4.5Ghz CPU and 16GB RAM, we set up a CA node, a contract node, and two peer nodes and use Ubuntu 20.04 as a server. To submit transaction data to the blockchain network and read data from the network, we developed a core chaincode to be a master operation. For this purpose, Calliper v0.4.2 was used as a performance measure to test the throughput and transaction latency of the Chaincode. The throughput of the blockchain network is the transaction commit rate on the entire ledger measured using TPS, and the latency is the time taken to test the interaction between the handover of the chaincode and the ledger. Transaction latency and throughput are used to perform read and write operations.

Before testing latency and throughput, we first tested memory and CPU usage without load balancing and with load balancing. Without load balancing, the blockchain receives requests for services in the order in which they're sent, which

**Table 4. Calculated costs per phase.**

| Organization Stage | CO | SCH | STU | IPFS |
|---|---|---|---|---|
| Company recruitment announcement phase | $T_{asy}+T_h+5T_{mul}$ | $T_{asy}+2T_{cmp}+T_h+6T_{mul}$ | N/A | $T_h$ |
| Student job application phase | N/A | $T_{asy}+2T_{cmp}+T_h+6T_{mul}$ | $T_{asy}+T_h+5T_{mul}$ | $T_h$ |
| Interview phase | $T_{asy}+T_{cmp}+4T_h+10T_{mul}$ | $2T_{asy}+4T_{cmp}+4T_h+12T_{mul}$ | $2T_{asy}+2T_{cmp}+2T_h+11T_{mul}$ | N/A |
| Announcement results phase | $T_{asy}+T_h+3T_{mul}$ | $T_{asy}+2T_{cmp}+T_h+4T_{mul}$ | N/A | N/A |

Notes:

$T_{asy}$: The time required for asymmetric signature/validation signature.

$T_{cmp}$: The time required to determine the operation.

$T_h$: Time required to calculate hash functions.

$T_{mul}$: Time required for multiplication operations.

**Table 5. Communication costs per phase.**

| Organization Stage | Message Length | Round | Fast Ethernet (100Mbps) | 10 Gigabit Ethernet (10 Gbps) |
|---|---|---|---|---|
| Company recruitment announcement phase | Tsig+5Tother= 512+5*80= 912 bits | 3 | 912/102400= 0.009 ms | 912/10240000= 0.09 us |
| Student job application phase | Tsig+2Tasy+3Tother= 512+2*1024+3*80= 2800 bits | 3 | 2800/102400= 0.027 ms | 2800/10240000= 0.27 us |
| Interview phase | 3Tsig+3Tasy+12Tother= 3*512+3*1024+12*80= 5568 bits | 6 | 5568/102400= 0.054 ms | 5568/10240000= 0.54 us |
| Announcement results phase | Tsig+3Tother= 512+3*80= 752 bits | 1 | 752/102400= 0.007 ms | 752/10240000= 0.07 us |

Notes:

$T_{sig}$: The time required to transmit the ECDSA signature (512 bits).

$T_{asy}$: Time required to send an asymmetric message (1024 bits).

$T_{other}$: The time required to send the message (80 bits)

reduces the response speed of blockchain nodes and takes up a lot of system resources, which can cause the server to crash. In this test, we set the transaction rate to 2000tps and continued the transaction for 2 seconds. In Figs 11 and 12, we see that without load balancing, the usage of the CPU is 60.56% and the memory usage is 538.7 MB. With load balancing, the usage of the CPU drops to 53.95%, and the memory usage drops to536.9 MB. In this test, we found that load balancing can optimize CPU utilization and memory usage.

In Fig 13, we analyze the relationship between the sending rate and the throughput. In this test, we chose 13 sending rate groups from 50 tps to 650 tps, each with 50 tps intervals of 3 seconds, and use load balancing. The throughput is 31.1 tps at a minimum and 124.8 tps at maximum for the write office and 50 tps at a minimum and 636.9 tps at maximum for the read office.

In Fig 14 we analyze the relationship between sending rate and latency. The minimum delay when writing data is 0.28 s and the maximum delay is 11.95 s. The minimum delay when querying data is 0.01 s and the maximum delay is 1.49 s. It can be seen that when the transmission interval size is constant, the delay of the writing phase increases with the increase of transmission rate, while the delay of the read phase does not increase with the increase of transmission rate, and the delay time is stable and short. The delay of the read phase before 550 tps does not increase with the increase of

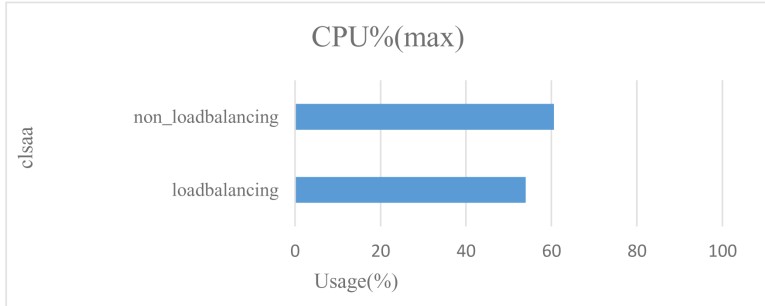

**Fig 11. Comparison of CPU consumption with and without load balancing.**

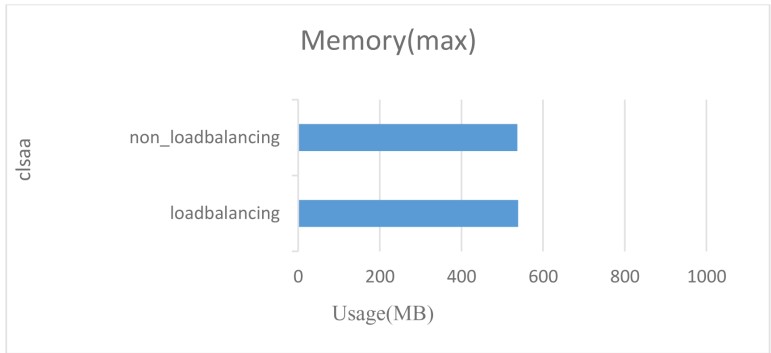

**Fig 12. Comparison of Memory consumption with and without load balancing.**

transmission rate, and the delay time is stable and short, but after 550 tps, the delay time increases with the increase of transmission rate. Therefore, the proposed system is very powerful in storing and reading data and can quickly publish information about the application in the application phase of the company. In addition, the throughput is large enough to write information to the CV and learning history files in the student's application phase.

### 5.4 Related Works Comparison

In this section, we conducted a literature review [8,13,15–19] and identified several shortcomings: [15–18] failed to ensure data integrity; [8,13,15,17–19] neglected the non-repudiation security requirement; [8,13,15–17] lacked access control mechanisms to limit access to sensitive data; [8,13,15] did not address scalability challenges; [8,13,15–19] did not employ off-chain storage technology on the blockchain to store data. In response, this study puts forth a talent recruitment system with enhancements to privacy, data integrity, and non-repudiation: leveraging cryptography to secure information transmission, mitigating blockchain storage limitations and facilitating company-university recruitment pipelines via off-chain storage, and enforcing access control policies to restrict unauthorized access to sensitive student data.

 Furthermore, we conducted a systematic comparison and evaluation of previous related research. Table 6 provides a detailed comparison of key features between our study and existing literature. Chen et al. [10] proposed using Hyperledger Fabric to solve the cross-university credit-credit certification system. Jeong and Choi [15] proposed a blockchain-
based digital certificate recruitment management platform. Guo et al. [16] proposed an intelligent graduate recruitment

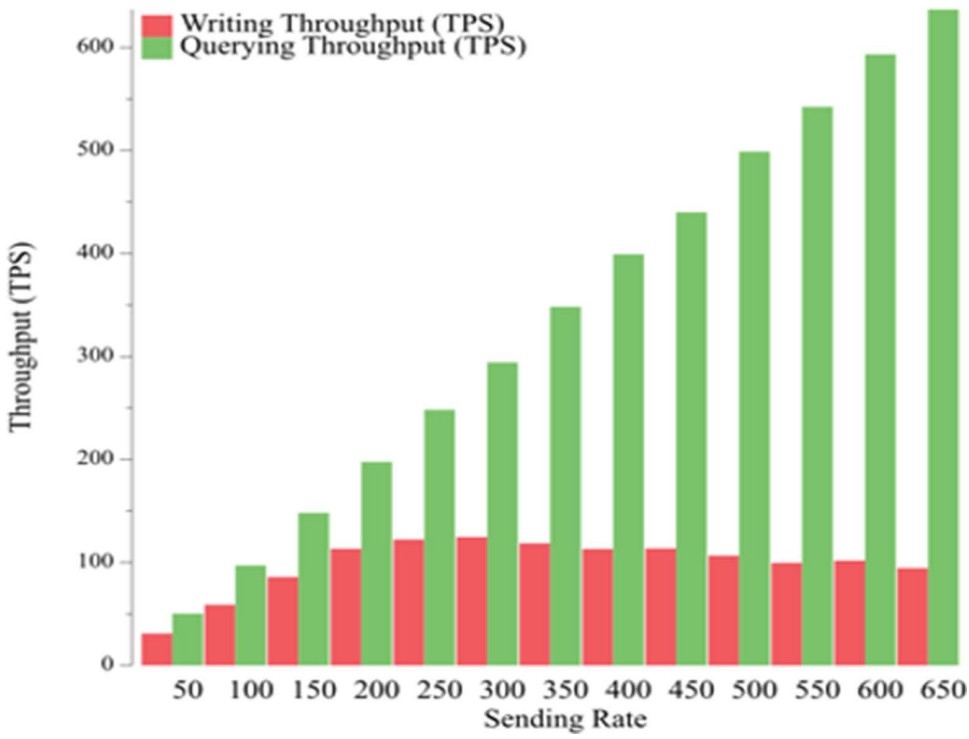

**Fig 13. Throughput in Hyperledger Fabric environment under different workloads.**

system based on a data-assisted blockchain. Hamrani et al. [30] proposed a disability recruitment model based on blockchain and smart contracts, but it increases the burden on the amount of on-chain storage and data owners. Turkanović et al. [31]proposed a blockchain-based educational record storage and sharing scheme that allowed students to transfer credits between different educational institutions. Still, it increases the amount of on-chain storage and the burden on data owners. Li and Han [32] proposed using Ethereum smart contracts to enable cross-institutional sharing of education records, but no external agency, only school-to-school. Ayub Khan et al. [33] proposed a degree-proof system based on Hyperledger Fabric traceability.

## 6 Conclusions

This study proposes a privacy-aware talent recruitment system based on the consortium blockchain and IPFS, which utilizes the non-tamperable mechanism of blockchain, IPFS, and asymmetric encryption to achieve data privacy and security in the talent recruitment management process.

The primary contributions of this study are as follows:

1) The immutability of blockchain prevents altering students' job search information, ensuring data integrity.

2) Utilizing the features of Hyperledger Fabric, the system mitigates the participation of malicious entities, thus reducing the risk of fraudulent information.

3) Implementing IPFS to store company recruitment data and students' job search data reduces on-chain data storage costs.

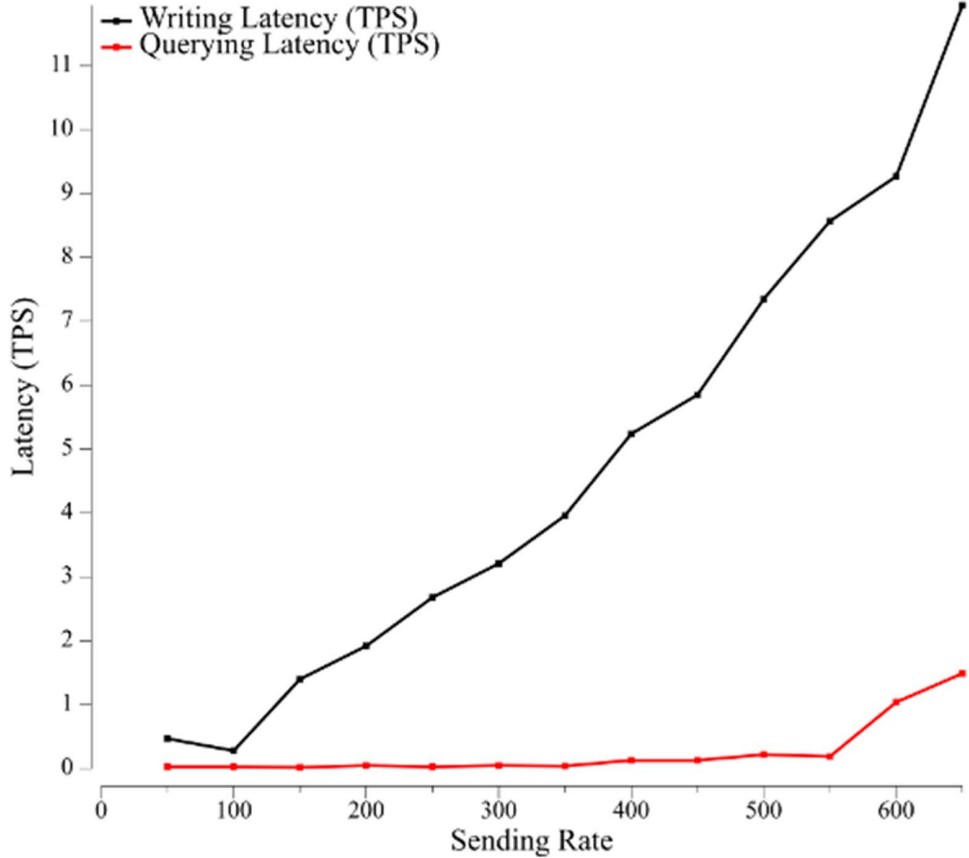

**Fig 14. Latency time in Hyperledger Fabric environment under different workloads.**

**Table 6. The proposed protocol was compared with the characteristics of the related works.**

| Author | Target | 1 | 2 | 3 | 4 | 5 | 6 | 7 |
|---|---|---|---|---|---|---|---|---|
| Chen et al. [10] | Using Hyperledger Fabric to Solve Inter-University Credit Verification System | Y | Y | N | N | N | N | N |
| Jeong and Choi. [15] | Digital Certificate Recruitment Management Platform Using Blockchain Technology | Y | Y | N | N | N | N | N |
| Guo et al.[16] | Smart Recruitment System for Graduates Using Big Data-Assisted Blockchain | Y | N | N | N | N | N | N |
| Hamrani and Hamrani. [30] | Recruiting people with disabilities using blockchain and smart contracts | Y | N | Y | N | Y | N | N |
| Turkanović et al. [31] | Using blockchain system to record the credits taken by university students to achieve secure and private storage of education records and sharing of education records. | Y | N | N | N | Y | N | N |
| Li and Han. [32] | Using blockchain to protect educational history files | Y | N | N | Y | Y | N | N |
| Ayub Khan et al. [33] | Using Hyperledger Fabric to improve security and privacy-related issues in the record traceability framework of the HEC degree certification system | Y | Y | N | Y | Y | Y | N |
| Ours | Propose a Hyperledger Fabric and IPFS-based system with privacy for talent recruitment | Y | Y | Y | Y | Y | Y | Y |

Notes:1: Blockchain architecture;2: Data integrity;3: Non-repudiation;4. With access control;

5: Scalability;6: Off-chain storage;7: School-company cooperation, (Y):Yes,(N):No.

4) Asymmetric encryption is employed to secure students' job applications, enhancing data privacy and security.

5) A consortium blockchain is established between schools and companies to facilitate secure information exchange, balancing data sharing and privacy protection.

6) The system provides a matchmaking platform for enterprises, schools, and students, ensuring secure and efficient employment opportunities.

Overall, the proposed scheme offers a secure and private job search experience for students while providing companies with a novel recruitment option. Companies can evaluate candidates by reviewing their academic history and performance, thereby improving the recruitment process.

In the future, we plan to integrate AI tools as another component to expedite and enhance the accuracy of the recruitment process in the human resources department. Additionally, we will determine the impact of node quantity and workload on the overall performance of the network

## Acknowledgments

We extend our appreciation to express our gratitude to all participants for their time devoted to this study.

## Author contributions

**Conceptualization:** Chin-Ling Chen, Kuang-Wei Zeng, Ling-Chun Liu.

**Formal analysis:** Hsing-Chung Chen.

**Investigation:** Hsing-Chung Chen, Yong-Yuan Deng.

**Methodology:** Chin-Ling Chen, Kuang-Wei Zeng, Ling-Chun Liu.

**Software:** Yong-Yuan Deng.

**Validation:** Yong-Yuan Deng, Chin-Feng Lee, Der-Chen Huang.

**Visualization:** Der-Chen Huang.

**Writing – original draft:** Kuang-Wei Zeng, Ling-Chun Liu.

**Writing – review & editing:** Chin-Ling Chen, Hsing-Chung Chen, Chin-Feng Lee.

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
