## [Decision Letter · Decision Letter 0]

18 Jul 2024

PONE-D-24-15807A Decentralized Talent Recruitment System with Resume Traceability and Privacy Protection Using Consortium Blockchain and IPFSPLOS ONE

Dear Dr. Chen,

Thank you for submitting your manuscript to PLOS ONE. After careful consideration, we feel that it has merit but does not fully meet PLOS ONE’s publication criteria as it currently stands. Therefore, we invite you to submit a revised version of the manuscript that addresses the points raised during the review process.

We look forward to receiving your revised manuscript.

Kind regards,

Fredrick Romanus Ishengoma

Academic Editor

PLOS ONE

Journal Requirements:

"This work was supported by the Chelpis Quantum Tech Co., Ltd., Taiwan, under the Grant number of Asia University: I112IB120. This work was supported by the National Science and Technology Council (NSTC), Taiwan, under NSTC Grant numbers: 112-2410-H-324 -001 -MY2, 111-2218-E-468-001-MBK, 110-2218-E-468-001-MBK, 110-2221-E-468-007, 111-2218-E-002-037 and 110-2218-E-002-044."

"This work was supported by the Chelpis Quantum Tech Co., Ltd., Taiwan, under the Grant number of Asia University: I112IB120. 

This work was supported by the National Science and Technology Council (NSTC), Taiwan, under NSTC Grant numbers: 112-2410-H-324 -001 -MY2, 111-2218-E-468-001-MBK, 110-2218-E-468-001-MBK, 110-2221-E-468-007, 111-2218-E-002-037 

and 110-2218-E-002-044."

"This work was supported by the Chelpis Quantum Tech Co., Ltd., Taiwan, under the Grant number of Asia University: I112IB120. This work was supported by the National Science and Technology Council (NSTC), Taiwan, under NSTC Grant numbers: 112-2410-H-324 -001 -MY2, 111-2218-E-468-001-MBK, 110-2218-E-468-001-MBK, 110-2221-E-468-007, 111-2218-E-002-037 and 110-2218-E-002-044."

6. We note that your Data Availability Statement is currently as follows: All relevant data are within the manuscript and its Supporting Information files.

7. When completing the data availability statement of the submission form, you indicated that you will make your data available on acceptance. We strongly recommend all authors decide on a data sharing plan before acceptance, as the process can be lengthy and hold up publication timelines. Please note that, though access restrictions are acceptable now, your entire data will need to be made freely accessible if your manuscript is accepted for publication. This policy applies to all data except where public deposition would breach compliance with the protocol approved by your research ethics board. If you are unable to adhere to our open data policy, please kindly revise your statement to explain your reasoning and we will seek the editor's input on an exemption. Please be assured that, once you have provided your new statement, the assessment of your exemption will not hold up the peer review process.

8. We note you have included a table to which you do not refer in the text of your manuscript. Please ensure that you refer to Table 6 in your text; if accepted, production will need this reference to link the reader to the Table.

**Additional Editor Comments:**

Comments from PLOS Editorial Office: We note that one or more reviewers has recommended that you cite specific previously published works. As always, we recommend that you please review and evaluate the requested works to determine whether they are relevant and should be cited. It is not a requirement to cite these works. We appreciate your attention to this request.

Reviewers' comments:

Reviewer's Responses to Questions

**Comments to the Author**

1. Is the manuscript technically sound, and do the data support the conclusions?

Reviewer #1: Partly

Reviewer #2: Partly

Reviewer #3: Partly

2. Has the statistical analysis been performed appropriately and rigorously? 

Reviewer #1: N/A

Reviewer #2: I Don't Know

Reviewer #3: Yes

3. Have the authors made all data underlying the findings in their manuscript fully available?

Reviewer #1: Yes

Reviewer #2: Yes

Reviewer #3: Yes

4. Is the manuscript presented in an intelligible fashion and written in standard English?

Reviewer #1: No

Reviewer #2: Yes

Reviewer #3: Yes

5. Review Comments to the Author

Reviewer #1: The paper presents the growing issue of unemployment among teenagers and college graduates and the challenges companies face in talent acquisition. Students face the risk of privacy breaches during job applications, while companies deal with high costs, inefficiency, and security concerns related to previous experiences. To address these issues, the authors propose using consortium chain technology and the InterPlanetary File System (IPFS) to create a decentralized talent recruitment system that allows encrypted data to be securely uploaded and accessed on the blockchain, mandating student authorization for resume access. The proposed system meets different blockchain requirements and performs well in several aspects, such as communication cost, computing cost, throughput, and transaction delay.

First of all, I would like to admit that the paper presents a useful application of consortium chain technology in the talent acquisition process, and good implementation can address the existing challenges to some extent.

However, the current shape of the paper, its readability, and the level of contributions are not strong enough to make the paper acceptable for publishing.

To be more precise:

- The research objective and research questions were not clearly defined.

- The paper doesn’t discuss real-life applications of blockchain technology to address similar issues in other fields.

- The study could profit from stakeholders’ opinions to understand the challenges faced as well as the needs of the system.

- The paper is poorly structured and needs thorough textual revisions. Sometimes, it is challenging to understand what the main value/novelty added.

Therefore, considering the reasons mentioned above, I recommend rejecting the paper in its current form.

Additional comments

- Title: Don't use abbreviations in the title

- Abstract: Refine the structure of the abstract.

- Introduction: Refine the structure of. Add statistics about the implications of unemployment and the effect of forging certificates on companies/government/society. Add a clear argument as to why we need blockchain. Clearly state the objective of the study and separate it from the methodology. For the section description, clearly state the topics to be discussed in Section 2.

- Proposed scheme: Add a short paragraph to about the elements of the system architecture.

- Notation: Provide a short paragraph to explain the notation are from where or what.

- Initialization phase: If this is a detailed description of the steps mentioned in the figure, then it is better to move it or merge it with the section above. or rename it as a different section. It seems random. Also, the size of the font is wrong for this subsection.

- Registration phase: as per the comment above.

- Company Recruitment Announcement Phase: same as the comment above.

- Student Job Application Phase: same as the comment above.

- Announcement Results Phase: same as the comment above.

- Analysis: provide a short paragraph introducing the section.

- Verifiable: “This paper achieves several verifiable goals through the architecture of the blockchain”. The sentence is confusing. Are the goals of the paper verifiable through the use of blockchain, or is the feature of verifiability achieved through blockchain?

- Man-in-the-middle Attack (MITM) and Sybil Attack can be merged under one section discussing cyber attacks. Also, why are only these two attacks discussed?

- Discussion: add a short paragraph describing the performance measures or KPIs you will discuss.

- Related Works Feature Comparison: the sentences are not well connected. needs restructuring.

- Conclusion: make sure to indicate the contribution in a paragraph. Add limitations and future work. Remove the duplicated paragraph.

Furthermore, thorough textual revision is needed. It contains many errors and typos.

Reviewer #2: In this manuscript, the authors proposed A Decentralized Talent Recruitment System with Resume Traceability and Privacy Protection Using Consortium Blockchain and IPFS

• This paper is an overall good.

• Implementation section is week, Authors need to include more results to validate the proposed approach and also include comparative analysis of the proposed approach with the existing approaches.

• The presented manuscript is well presented but lack supportive references. In fact, one should expect a reasonable number of references in order to support the claims by literature.

Some of the latest work can be cited as below:

• Introduction to Data Analytics ( https://doi.org/10.4018/979-8-3693-3609-0.ch001)

• Data Ethics and Privacy (https://doi.org/10.4018/979-8-3693-3609-0.ch011)

• ArMor: A data analytics scheme to identify malicious behaviors on blockchain-based smart grid system

• A data analytics scheme for security-aware demand response management in smart grid system

• RAKSHAK: Resilient and scalable demand response management scheme for smart grid systems

Reviewer #3: The proposed work is good, but it needs the following revisions

1- The paper is quite important and relevant, but I found the technical contribution of this paper is needed to be relooking.

2- Title is needed to relook and make it more appropriate in the view of contribution.

3- Further, The paper's abstract should focus on the importance of addressing this issue. The highlights of the results and how the reader might benefit from the paper's material should also be included.

4- The Introduction section needs to relook to improve its quality and readability.

5- It is also suggested to elaborate the mathematical term in details.

6- Further¸ it is suggested in Related works Section, need to add the existing work and make a comparative table to address the various key gaps in existing work which are going to address through this paper. There is not much recent literature is included in this manuscript, some of the recent literature are listed below. Rather than some other related works are there can be included:

https://www.mdpi.com/2076-3417/11/14/6376

https://www.techscience.com/cmc/v74n3/50945

https://www.sciencedirect.com/science/article/abs/pii/S2542660523002925

https://ieeexplore.ieee.org/abstract/document/10189750

https://ieeexplore.ieee.org/abstract/document/10584093

https://www.mdpi.com/2076-3417/13/2/691

https://www.mdpi.com/2224-2708/11/3/55

https://ieeexplore.ieee.org/abstract/document/7502325

https://ieeexplore.ieee.org/abstract/document/7492748

https://ieeexplore.ieee.org/abstract/document/7490948

7- Authentication phase for (3.6 Student Job Application Phase) and for the proposed protocol need to be relooked and explain in more details.

8- Justify the security analysis to offer some formal security analysis of the proposed scheme.

9- Further the used Sample and Procedure need to be validating in the respect of measurements of Communication overheads comparisons.

10- Include a brief statement about possible applications that could profit from the suggested methodology in the conclusions section.

6. PLOS authors have the option to publish the peer review history of their article (what does this mean? ). If published, this will include your full peer review and any attached files.

**Do you want your identity to be public for this peer review?** For information about this choice, including consent withdrawal, please see our Privacy Policy .

Reviewer #1: No

Reviewer #2: No

Reviewer #3: No

---

## [Author Response · Author response to Decision Letter 1]

30 Jul 2024

Manuscript ID: PONE-D-24-15807

Reviewer 1 comments: The paper presents the growing issue of unemployment among teenagers and college graduates and the challenges companies face in talent acquisition. Students face the risk of privacy breaches during job applications, while companies deal with high costs, inefficiency, and security concerns related to previous experiences. To address these issues, the authors propose using consortium chain technology and the InterPlanetary File System (IPFS) to create a decentralized talent recruitment system that allows encrypted data to be securely uploaded and accessed on the blockchain, mandating student authorization for resume access. The proposed system meets different blockchain requirements and performs well in several aspects, such as communication cost, computing cost, throughput, and transaction delay.

First of all, the paper presents a useful application of consortium chain technology in the talent acquisition process, and good implementation can address the existing challenges to some extent.

However, the current shape of the paper, its readability, and the level of contributions are not strong enough to make the paper acceptable for publishing.

To be more precise:

1. The research objective and research questions were not clearly defined.

Author’s Response:

Thanks for the reviewer's valuable suggestion of the study. To clarify, as delineated in our abstract, our research question addresses the increasingly critical issue of youth unemployment, focusing on the challenges college graduates face in securing 00employment post-graduation. Concurrently, we examine the multifaceted obstacles companies encounter in their talent acquisition processes, including but not limited to high operational costs, security vulnerabilities, procedural inefficiencies, and time-intensive sourcing methodologies. Furthermore, our study acknowledges the risks confronted by job applicants, specifically concerning the potential compromise of personal data during the application process.

With respect to our research objectives, as articulated in the abstract, our study proposes a novel approach utilizing consortium chain technology in conjunction with the InterPlanetary File System (IPFS) to develop a decentralized talent recruitment system. This approach enables students, educational institutions, and potential employers to encrypt and upload data to the blockchain through consortium chain technology, with strict access controls requiring student authorization for resume data retrieval. The proposed system facilitates a symbiotic relationship between educational institutions and industry partners, allowing students to identify suitable employment opportunities while enabling companies to source candidates with requisite expertise efficiently.

2. The paper doesn’t discuss real-life applications of blockchain technology to address similar issues in other fields.

Author’s Response:

Thanks for the reviewer's valuable suggestion of the study. In response to this constructive feedback, we have augmented the third paragraph of our Introduction section with additional literature on blockchain technology's application in addressing challenges within the healthcare insurance domain. Corresponding modifications have been implemented on page 2 of our manuscript to reflect these enhancements.

3. The study could profit from stakeholders’ opinions to understand the challenges faced as well as the needs of the system.

Author’s Response:

Thank you for the reviewer's valuable suggestion. In response, we have added a discussion on the integrity issues faced by graduates during job interviews in the second paragraph of the introduction. The corresponding modifications have been made on page 2.

4. The paper is poorly structured and needs thorough textual revisions. Sometimes, it is challenging to understand what the main value/novelty added.

Author’s Response:

Thanks for the reviewer's valuable suggestion of the study. This research leverages blockchain to transform centralized job-seeking systems into a decentralized architecture. The immutability of blockchain ensures the authenticity of students' course records and grades, preventing denial of their learning history. IPFS is utilized to reduce storage costs, while encryption techniques safeguard privacy. An alliance chain is established between schools and businesses to facilitate information sharing. This platform provides a secure and reliable matching mechanism for companies, educational institutions, and students, aiding in the identification of suitable talent. The decentralized approach enhances trust and efficiency in the job market, offering a robust talent acquisition and career development solution.

5. Title: Don't use abbreviations in the title

Author’s Response:

Thanks for the reviewer's valuable suggestion of the study. We have made the corresponding modifications on page 1.

6. Abstract: Refine the structure of the abstract.

Author’s Response:

We extend our sincere gratitude for the insightful comments provided by the reviewers. We have made corresponding modifications to the abstract on page 1.

7. Introduction: Refine the structure of. Add statistics about the implications of unemployment and the effect of forging certificates on companies/government/society. Add a clear argument as to why we need blockchain. Clearly state the objective of the study and separate it from the methodology. For the section description, clearly state the topics to be discussed in Section 2.

Author’s Response:

Thanks for the reviewer's valuable suggestion of the study. we have comprehensively revised and enhanced the introduction section on pages one and two. The specific modifications are as follows:

1. Structural Optimization:We have meticulously restructured the introduction to enhance its clarity and logical flow, ensuring a more coherent presentation of our research context and objectives.

2. Statistical Data Augmentation:We have incorporated pertinent data on the societal impact of unemployment. For instance, we now highlight that unemployment significantly increases the risk of social exclusion, potentially leading to long-term negative consequences for both individuals and society at large.

3. A new paragraph has been added to the third section, elucidating the rationale behind employing blockchain technology in this study. We emphasize the decentralized, anonymous, traceable, and non-repudiable characteristics of blockchain and their potential to address prevalent issues in talent recruitment, such as fraudulent recruitment practices and data breaches.

4. Research Objectives Clarification: We have explicitly articulated our research objectives, distinguishing them from the methodological approach to provide a clearer roadmap for the study.

5. Preview of Chapter Two: In the concluding part of the introduction, we have expanded on the challenges faced by graduates during interviews. Specifically, we address the integrity issues that may arise during the recruitment process as candidates strive to appear more attractive to potential employers.

We are confident that these revisions have substantially enhanced the quality and readability of the introduction, laying a robust foundation for the entire manuscript. The modified introduction now provides a more comprehensive context, clearer objectives, and a stronger justification for our research approach.

8. Proposed scheme: Add a short paragraph to about the elements of the system architecture.

Author’s Response:

Thanks for the reviewer's valuable suggestion of the study. we have augmented our manuscript with an additional elucidation of the system architecture's components in the latter part of page 4. The supplementary text reads as follows:

Section 3.1, System Architecture, delineates the structural framework of the proposed system, providing a comprehensive exposition of the various roles, systems, and organizations involved. This section culminates in presenting0 the complete process flow, encompassing the spectrum from disseminating recruitment information to the student-employer interview phase. Subsequently, section 3.2, Notation, introduces and defines all symbols and notations utilized throughout the subsequent 0discourse. Sections 3.3 through 3.8 offer a more granular examination of each phase in the recruitment process, from the dissemination of job postings to the actualization of student-employer interviews, detailing the specific protocols and communication mechanisms employed at each juncture.

9. Notation: Provide a short paragraph to explain the notation are from where or what.

Author’s Response:

Thanks for the reviewer's valuable suggestion of the study. The description of the notation has been updated in the relevant section on page 5.

10. Initialization phase: If this is a detailed description of the steps mentioned in the figure, then it is better to move it or merge it with the section above. or rename it as a different section. It seems random. Also, the size of the font is wrong for this subsection.

Author’s Response:

Thanks for the reviewer's valuable suggestion of the study. We have made the corresponding modifications on page 5 and page 6.

11. Registration phase: as per the comment above.

Author’s Response:

Thanks for the reviewer's valuable suggestion of the study. We have made the corresponding modifications on page 6.

12. Company Recruitment Announcement Phase: same as the comment above.

Author’s Response:

Thanks for the reviewer's valuable suggestion of the study. We have made the corresponding modifications on page 7.

13. Student Job Application Phase: same as the comment above.

Author’s Response:

Thanks for the reviewer's valuable suggestion of the study. We have made the corresponding modifications on page 8.

14. Announcement Results Phase: same as the comment above.

Author’s Response:

Thanks for the reviewer's valuable suggestion of the study. We have made the corresponding modifications on page 11 and page 12.

15. Analysis: provide a short paragraph introducing the section.

Author’s Response:

Thanks for the reviewer's valuable suggestion of the study. We have addressed this by adding a short introductory paragraph for the subsequent sections on page 12. This addition provides a concise overview of the content, enhancing the paper's structure and readability.

16. Verifiable: “This paper achieves several verifiable goals through the architecture of the blockchain”. The sentence is confusing. Are the goals of the paper verifiable through the use of blockchain, or is the feature of verifiability achieved through blockchain?

Author’s Response:

Thanks for the reviewer's valuable suggestion of the study. We agree that the original sentence was ambiguous. We have revised it to: " This paper implements several verifiable goals in the system using blockchain architecture" This clarification better reflects our intention to showcase how blockchain technology enables system-wide verifiability for key aspects like identity authentication and transaction transparency. The revised statement appears on page 14 of the manuscript.

17. Man-in-the-middle Attack (MITM) and Sybil Attack can be merged under one section discussing cyber attacks. Also, why are only these two attacks discussed?

Author’s Response:

Thanks for the reviewer's valuable suggestion of the study. Our focus on Man-in-the-Middle (MITM) and Sybil attacks stems from their significant prevalence and impact in both traditional network and blockchain environments. These attack methods represent critical vulnerabilities extensively documented in the literature. This approach allows for a comprehensive analysis of core attack mechanisms and defense strategies, providing valuable insights for both conventional and emerging network paradigms.

18. Discussion: add a short paragraph describing the performance measures or KPIs you will discuss.

Author’s Response:

Thanks for the reviewer's valuable suggestion of the study. We have incorporated an additional paragraph on page 14 to elucidate the performance evaluation discussed in Chapter 5. The supplementary text is as follows:

Section 5.1 of this chapter will analyze the computational costs for each phase. Section 5.2 will examine the communication costs for each phase, evaluating them in both Fast Ethernet and 10 Gigabit Ethernet environments. Section 5.3 will present performance tests based on the proposed solution, focusing on the performance evaluation of chaincode deployment. In the test, we selected 13 sending rate groups ranging from 50 tps to 650 tps, with 50 tps intervals between each group, each lasting 3 seconds, and employing load balancing techniques. Furthermore, we will analyze the relationship between sending rate and latency, as well as the relationship between sending rate and throughput. Finally, we will analyze the access capability and throughput of the proposed system. Section 5.4 will compare our approach with previous research using a tabular format, considering the following seven aspects: 1) Blockchain architecture; 2) Data integrity; 3) Non-repudiation; 4) Access control; 5) Scalability; 6) Off-chain storage; 7) School-company cooperation.

19. Related Works Feature Comparison: the sentences are not well connected. needs restructuring.

Author’s Response:

Thanks for the reviewer's valuable suggestion of the study. We have made corresponding modifications to the Related Works Feature Comparison on page 17.

20. Conclusion: make sure to indicate the contribution in a paragraph. Add limitations and future work. Remove the duplicated paragraph.

Author’s Response:

Thank you for the reviewer's valuable suggestions regarding our study. We have made the corresponding revisions on page 18 and removed the redundant paragraphs.

Reviewer 2 comments:

In this manuscript, the authors proposed A Decentralized Talent Recruitment System with Resume Traceability and Privacy Protection Using Consortium Blockchain and IPFS

1. This paper is good overall.

Author’s Response:

Thanks for the reviewer's positive comments.

2. The Implementation section needs to be stronger; authors need to include more results to validate the proposed approach and also include a comparative analysis of the proposed approach with the existing approaches.

Author’s Response:

Thanks for the review's comments. We want to draw attention to the third paragraph of our Introduction, which presents a comprehensive exploration of existing methodologies. Furthermore, Section 5.4 of our manuscript provides a comparative analysis between our proposed approach and current methods in the field. This juxtaposition serves to contextualize our contribution within the existing body of literature and elucidate the advancements offered by our research.

3. The presented manuscript is well presented but lack supportive references. In fact, one should expect a reasonable number of references in order to support the claims by literature.

Some of the latest work can be cited as below:

• Introduction to Data Analytics ( https://doi.org/10.4018/979-8-3693-3609-0.ch001)

• Data Ethics and Privacy (https://doi.org/10.4018/979-8-3693-3609-0.ch011)

• ArMor: A data analytics scheme to identify malicious behaviors on blockchain-based smart grid system

• A data analytics scheme for security-aware demand response management in smart grid system

• RAKSHAK: Resilient and scalable demand response management scheme for smart grid systems

Author’s Response:

Thanks for the reviewer's valuable suggestion of the study. We have incorporated the relevant studies into our references [20] and [21] and made corresponding revisions on page 2 of the introduction. These additions enhance the context and support for our research, aligning it more closely with existing literature in the field.

4. In the implementation section, authors may give some explanation regarding hyper ledger fabric and IPFS part.

Author’s Response:

Thank you for the reviewer's valuable suggestion. In the implementation phase, we introduce Hyperledger Fabric. Section 2 provides a clearer explanation of how Hyperledger Fabric and IPFS operate together. IPFS serves as the blockchain's storage system. After storing data in IPFS and obtaining a Content Identifier

---

## [Decision Letter · Decision Letter 1]

25 Nov 2024

Secure and Efficient Graduate Employment: A Consortium Blockchain Framework with InterPlanetary File System for Privacy-Preserving Resume Management and Efficient Talent-Employer Matching

PONE-D-24-15807R1

Dear Dr. Chen,

We’re pleased to inform you that your manuscript has been judged scientifically suitable for publication and will be formally accepted for publication once it meets all outstanding technical requirements.

Kind regards,

Shadab Alam, Ph.D.

Academic Editor

PLOS ONE

Additional Editor Comments (optional):

Reviewers' comments:

Reviewer's Responses to Questions

**Comments to the Author**

1. If the authors have adequately addressed your comments raised in a previous round of review and you feel that this manuscript is now acceptable for publication, you may indicate that here to bypass the “Comments to the Author” section, enter your conflict of interest statement in the “Confidential to Editor” section, and submit your "Accept" recommendation.

Reviewer #2: All comments have been addressed

Reviewer #4: All comments have been addressed

2. Is the manuscript technically sound, and do the data support the conclusions?

Reviewer #2: Yes

Reviewer #4: Partly

3. Has the statistical analysis been performed appropriately and rigorously? 

Reviewer #2: Yes

Reviewer #4: Yes

4. Have the authors made all data underlying the findings in their manuscript fully available?

Reviewer #2: Yes

Reviewer #4: Yes

5. Is the manuscript presented in an intelligible fashion and written in standard English?

Reviewer #2: Yes

Reviewer #4: Yes

6. Review Comments to the Author

Reviewer #2: A thorough proofreading/restructuring/grammar/sentence formation and spelling checking of this article is essential.

Reviewer #4: Manuiscript is well written and presented a research in BVlockchain with Hyperledger fabric. Data provided for the results and comparisions are sufficient. Integration of Consortium Blockchain and IPFS: They have developed a novel

framework that combines the strengths of consortium blockchain for secure, tamper-proof record-keeping with IPFS for efficient, decentralized data storage. This integration addresses both data integrity and storage efficiency challenges in

graduate employment systems.

7. PLOS authors have the option to publish the peer review history of their article (what does this mean? ). If published, this will include your full peer review and any attached files.

**Do you want your identity to be public for this peer review?** For information about this choice, including consent withdrawal, please see our Privacy Policy .

Reviewer #2: No

Reviewer #4: No

---

## [Editor Report · Acceptance letter]

PONE-D-24-15807R1

PLOS ONE

Dear Dr. Chen,

I'm pleased to inform you that your manuscript has been deemed suitable for publication in PLOS ONE. Congratulations! Your manuscript is now being handed over to our production team.

Kind regards,

on behalf of

Dr. Shadab Alam

Academic Editor

PLOS ONE